# Serious game versus standard care for rehabilitation after distal radius fractures: a protocol for a multicentre randomised controlled trial

Henriëtte A W Meijer ,[1] Maurits Graafland,[2] Miryam C Obdeijn,[3] Susan van Dieren,[2] J Carel Goslings,[4] Marlies P Schijven ,[5] on behalf of The ReValidate! Collaborative Study Group

► Prepublication history and additional materials for this paper are available online. To view these files, please visit the journal online (http://dx.doi.org/10.1136/bmjopen-2020-042629).

For numbered affiliations see end of article.

**Correspondence to**
Professor Marlies P Schijven;
m.p.schijven@amsterdamumc.nl

## ABSTRACT

**Introduction** Distal radius fractures are among the most prevalent traumatic injuries worldwide. These injuries are associated with high healthcare-related and socioeconomic costs, mainly resulting from loss of productivity. To optimise recovery and return to work, wrist exercises are recommended. However, adherence to standard exercise regimens is low. Serious games provide a treatment platform for standardised postoperative care, uniting meaningful recovery with entertainment. Also, mobile serious games, for example, smartphone or tablet applications, are able to send practice reminders believed to improve self-efficacy.

**Methods and analysis** To test the effectiveness of a mobile serious game for distal radius fracture rehabilitation compared with standard care, a multicentre, randomised controlled clinical trial was designed. Primary outcome will be the Patient-Rated Wrist Evaluation (PRWE) score after 6 weeks of treatment. Secondary outcomes are range of motion, grip strength, pain scores, and self-reported treatment adherence after 2, 6 and 12 weeks of treatment. Adult patients with any type of closed distal radius fracture are included directly after non-operative or operative fracture treatment. Patients are recruited in the outpatient clinics of four teaching hospitals. The intended sample size is 92 patients, based on the minimal clinically important difference of the PRWE score at 6 weeks, using a superiority model.

Patients are randomised between using the wearable-controlled mobile serious game *ReValidate!* (intervention group) and standard care consisting of unsupervised exercises and a referral for physiotherapy or exercise therapy upon request or recommendation by the treating clinician (control group).

**Ethics and dissemination** The protocol has been approved by the Medical Ethical Review Board of the Amsterdam University Medical Centres, location Academic Medical Centre in Amsterdam, the Netherlands. Results will be made available to involved healthcare providers, funders, and to the general public including patients via peer-reviewed academic journals and international conferences.

**Trial registration number** Dutch Trial Registry (NTR), NL6140, protocol V.2.

### Strengths and limitations of this study

► This study is a randomised controlled trial comparing a wearable-controlled mobile serious game for wrist rehabilitation with standard care, the first of its kind, to the authors' knowledge.

► The study is sufficiently powered to show a clinically important difference in the outcome measurements of the primary endpoint.

► By measuring treatment adherence, both subjectively and objectively, a reliable comparison between patient-reported and objectively monitored treatment adherence can be made.

► The control group receives an accurate representation of the current standard of care.

► The analysing statistician will be blinded to randomisation group.

## INTRODUCTION

Distal radius fractures are among the most frequently occurring traumatic injuries worldwide, occurring in 18% of patients who present to the emergency department after having sustained any fracture.[1–3] Up to 25% of fractures occurring in paediatric patients and 18% of fractures in the elderly population are distal radius fractures,[1 4] with an incidence of 30 and 25 per 10 000 person-years for the age groups under 18 and over 65 years old, respectively.[5] In the age group over 65 years old, the incidence in women is five times higher than that in men.[4 6] As the general population ages, the incidence of these injuries is increasing.[1 6–9] Not only are these injuries a burden to patients affected, they also present a significant socioeconomic burden to society. Hand and wrist injuries are expensive mainly due to the loss of productivity. Distal radius fractures form the largest and most expensive group within this population.[10–13]

Distal radius fractures can be treated either operatively or conservatively.[14] In order to speed up and enhance recovery of mobility and function after initial fracture treatment, patients are often referred for physiotherapy or hand therapy. Yet, currently available scientific literature remains inconclusive on whether supervised physiotherapy or unsupervised exercise programmes should be advocated.[15–18] There is consensus that starting exercises early is preferred over starting later,[19 20] and that performing any type of exercise is most likely more effective than performing no exercises.[21]

Similarly, surgical guidelines from various countries state that exercises are most likely beneficial.[22–25] These guidelines mostly leave it up to the clinician's preferences whether or not to refer a patient to a physiotherapist, or recommend a referral when patients suffer severe pain or report oedema.[15 22–25] This may lead to arbitrary rehabilitation regimens differing per region, hospital and even per individual healthcare provider. It is currently unknown what percentage of patients follow up to physiotherapy referrals. When patients are referred for physiotherapy and follow up to this referral, it has been found that only 19%–35% of prescribed exercise regimens are executed completely and correctly.[26] Treatment adherence is influenced by practical constraints such as time, costs and travel distances.[27] Patients with a low self-efficacy will also perceive more barriers to treatment adherence in a home-based exercise programme.[28] Self-efficacy is defined as the belief one can be successful when performing a certain task and is an important contributor to physiotherapy outcomes.[29 30]

In physiotherapy and rehabilitation, so-called 'serious games' have been gaining attention for their presumed effects on motivation and functional outcomes.[31–33] A serious game is defined as any kind of interactive computer application that incorporates gamification principles and serves an educational purpose, or aims to achieve a predefined goal.[34] Most studies evaluating the effect of serious games in physiotherapy use consumer-based game consoles with an external hardware component, for example, the Nintendo Wii.[35–37] These types of games enable patients to perform rehabilitation exercises. As a downside, such games require a game console and a monitor to function, limiting patients in their ability to exercise anywhere they want. A systematic review evaluating the use of serious games in rehabilitation of traumatic injuries described only 'off-the-shelf' games. No 'wearable-controlled' games could be identified.[32] Wearable sensors combined with easily accessible gaming platforms such as smartphones or tablets allow the user to exercise anytime and anywhere they want, and can decrease barriers to treatment adherence. To the best of our knowledge, there is no previous research on a home-based wearable-controlled serious game for traumatic injuries.

This multicentre, randomised controlled clinical trial aims to evaluate the effects on functional outcomes of a wearable-controlled serious game played on a smartphone or tablet, that is developed specifically for wrist rehabilitation. This is compared with the current standard treatment following distal radius fracture. This study hypothesises that the use of the game may lead to a higher treatment adherence by improving motivation and adding entertainment to exercises, thus increasing self-efficacy. It is therefore hypothesised that patients recovering with use of the game will show a greater improvement in patient-rated functional outcomes compared with standard treatment. Second, this study aims to compare active range of motion (ROM), grip strength, treatment adherence, pain scores and to measure the effects on return to work between the two treatment strategies.

## METHODS AND ANALYSIS
### Study design
The study is a two-arm, parallel-group, multicentre randomised controlled superiority trial designed to evaluate the effectiveness of a wearable-controlled serious game played on a smartphone or a tablet for distal radius fracture rehabilitation. Patients are randomised in a 1:1 ratio.

### Participants and setting
Consecutive patients aged 18 years or older, with any type of conservatively or operatively treated closed distal radius fracture, are eligible to participate in the study. Table 1 provides a complete overview of inclusion and exclusion criteria. Patients are recruited from four different teaching hospitals in the country, of which one is an academic centre. Patients are recruited at the outpatient clinics, where clinicians will be approached for permission to contact their patients about the study. Before starting trial participation, written informed consent will be obtained from all participants by the research staff.

Patients with a medical history of loss of function due to injury or illness in either hand or wrist, polytraumatised patients, as well as patients who suffered bilateral wrist fractures or other injuries to the affected limb, are excluded. Patients who are not in the possession of a smartphone or tablet compatible with the serious game, as well as patients with a visual or mental impairment preventing the proper use of smartphone or tablet, are excluded. Those unable to understand spoken or written Dutch or English will be unable to complete the questionnaires and are therefore not eligible for participation.

### Randomisation
When patients have been found eligible, and after they signed the informed consent form, baseline measurements are obtained. After baseline measurements, the local researcher contacts the central research coordinator for randomisation to take place and to receive information on group allocation. Randomisation sequence is determined by an independent researcher using an online automatically generated randomisation list (via www.sealedenvelope.com). Allocation concealment is

**Table 1** Inclusion and exclusion criteria

| Inclusion criteria | Exclusion criteria |
|---|---|
| 1. Any type of closed distal radius fracture with no injury to the skin of the affected limb | 1. Polytraumatised patients (Injury Severity Score ≥16) |
| 2. Good position after reposition or operative fixation, defined by <10° of inclination in any direction, <5 mm shortening of the radius compared with the ulna, <2 mm disposition of intra-articular fragments | 2. Bilateral wrist fractures or other concomitant injuries to the affected limb |
| 3. Fracture primarily treated with conservative (cast) immobilisation or operative fixation (ORIF) | 3. Patients with other disease or injury causing a clinically relevant loss of function or range of motion in the wrist, as reported by patients (including Parkinson's disease, having had a cerebral vascular accident, amyotrophic lateral sclerosis, neuropathy of any kind) |
| 4. Fracture considered to be consolidated by treating physician (trauma or orthopaedic surgeon or surgical resident in training) | 4. Previous fractures or any condition affecting the injured wrist with clinically relevant residual pain, loss of function or range of motion |
| 5. Possible to start rehabilitation exercises within 5 days after cast removal or operative fixation, as decided by treating physician (trauma or orthopaedic surgeon or surgical resident in training) | 5. Any medical contraindication to start rehabilitation within 5 days after operation or cast removal, including dislocation of the fracture, tendon rupture or complex regional pain syndrome, as decided by the treating specialist |
| 6. Age ≥18 years | 6. Insufficient proficiency of Dutch or English in speech and written language, or inability to complete the Dutch questionnaires |
| 7. Written informed consent | 7. Not in the possession of, or able to obtain for the duration of our study, a smartphone or tablet compatible with the serious game |
| | 8. Visual impairment preventing use of the smartphone-based game |

ORIF, open reduction internal fixation.

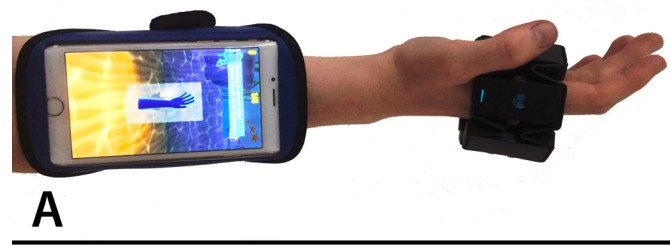

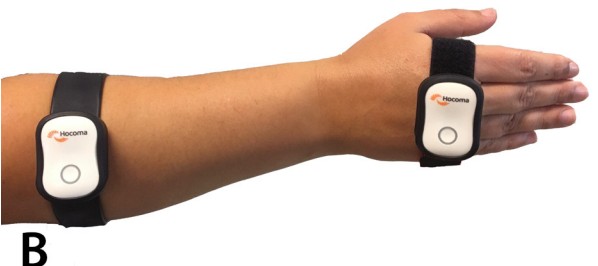

**Figure 1** Sensor placement. (A) iPhone and Myo; (B) two Valedo sensors.

## Interventions

After randomisation, patients start participation in the trial as soon as their fracture has been qualified as stable or consolidated, and patients are fit to start active exercises as determined by the clinician; this is either directly, or as soon as possible, but within 3–5 days after cast removal or successful operative fixation. Together with specialised hand and wrist physiotherapists and exercise therapists, a standardised protocol for the first 6 weeks—when most progress is to be expected—has been established that is both embedded within the game and explained to patients.[38] All patients are instructed by their clinician to perform the hand and wrist rehabilitation exercises three to five times per day. In addition, patients receive a simple diary to complete, with instructions to register their frequency and duration of exercises daily, as well as any physical complaints they may have during exercising.

### Intervention group

Patients who are randomised to the game group receive a download link via email, together with instructions on how to download and install the game on their own smartphone or tablet (iPad or iPhone, Apple, Cupertino, California, USA). Patients are instructed to strap one motion sensor to the dorsum of the hand and to strap a second one to the proximal forearm (figure 1). If patients are only in the possession of a compatible smartphone, they will receive the Myo gesture control armband (Thalmic Labs, Kitchener, Ontario, Canada). This sensor is worn around the hand, in combination with the patient's own smartphone strapped around the forearm with the screen upwards and still visible (figure 1). If patients are only in the possession of a compatible tablet, or if they prefer this option over using their smartphone, they receive

obtained using sealed opaque envelopes. Patients are randomised using block randomisation in blocks of four, stratified for age (18–64, and 65 years or above) and for treatment type (conservative treatment using immobilisation or operative treatment by means of internal fixation) to ensure a balanced randomisation into the two groups.

two Valedo sensors (Hocoma AG, Zurich, Switzerland). Two separate sensors are used since the tablet cannot be strapped to the arm to be used as a motion sensor. Both types of sensors control the exact same game in the same manner, the motions are the same, and gameplay in both types of game controller has been validated extensively, as described previously.[38] Both types of motion sensors are CE-marked, have been found fit to act as a game controller and are able to monitor wrist ROM. The sensors are connected to the smartphone via Bluetooth. By using two separate motion sensors placed proximally and distally of the wrist joint, isolated wrist motions are used for game control and patients cannot 'cheat' by moving the shoulder joint or the fingers. All otherwise necessary instructions are embedded within the game, making the game 'self-explanatory'. Intervention group patients will practise with the game for 6 weeks, after which they return the motion sensor to the hospital and are encouraged to continue exercising independently.

## Gameplay

The *ReValidate!* game shows an underwater world in which the patient plays one level consisting of three different mini-games of similar difficulty (online supplemental video). Each mini-game has its own avatar, and is controlled by one specific wrist motion: the 'anglerfish' mini-game is controlled by pronation and supination, the 'shark' mini-game is controlled by palmar and dorsal flexion of the wrist, and the 'penguin' mini-game is controlled by radial and ulnar deviation (figure 2). The patient plays one level three to five times per day, for a duration of approximately 10–15 min. Every playing session starts with a 'warm-up' of the motions, during which the motion sensors are calibrated and the game is set to the patients' own maximum ROM. This ensures the game remains challenging yet playable, and prevents overstraining the wrist to make movements outside of the patients' own ROM. A new level is unlocked each

day, provided that at least one playing session has been completed that day, so the game remains motivating and challenging. All levels consist of an underwater parkour that needs to be completed, only the surroundings and route change. Levels gradually increase in length and difficulty over the course of 6 weeks, in the same manner as a physiotherapy treatment programme increases in intensity over time. Patients are motivated by a daily 'push notification' reminder to exercise, and by optional high score rankings. Bonus points can be gathered by playing consistently three to five times per day and by obtaining 'collectibles' found in the different levels. The game application registers frequency and duration of gameplay and records progress in terms of ROM in degrees.

Because the swimming motions of the fish are comparable with the wrist motions needed to control the mini-games, gameplay through the embedded wrist exercises is intuitive and natural.

The game has been designed specifically for wrist rehabilitation by a company specialised in the development of rehabilitation games. An iterative design process was set up to ensure embedding of the standard exercise protocols. The game has been tested at all stages of development by experts and novices, including trauma surgeons, specialised hand and wrist surgeons, occupational therapists and physiotherapists, as well as by end-users.[38] Content has been checked for validity and exercise completeness both by user groups of patients and medical experts, and was found to be similar to regular physiotherapy exercise regimens.[38] Should patients encounter any problems with the game, they are provided contact details of a study coordinator. For any medical questions or concerns, patients are referred back to their treating specialist. Though not preferred, patients receive a referral for additional physiotherapy upon request. They will be motivated by their clinician to continue practising using the game. The number of physiotherapy visits will be registered prospectively.

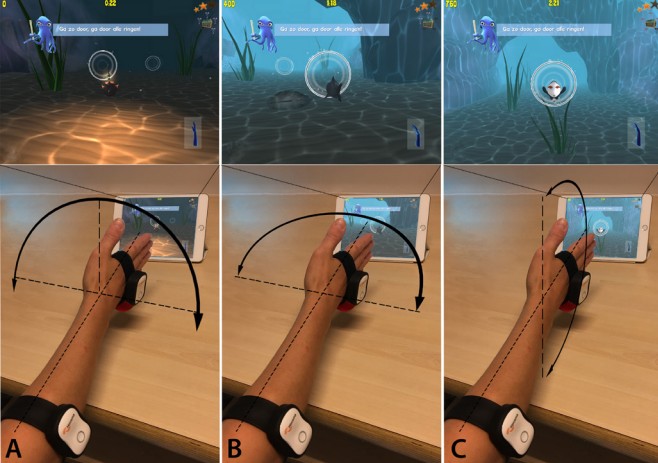

**Figure 2** Different avatars are used in the game. (A) The anglerfish is controlled by pronation and supination; (B) the shark is controlled by palmar and dorsal flexion; (C) the penguin is controlled by radial and ulnar deviation.

## Control group

Patients in the control group receive standard care, consisting of home-based unsupervised wrist exercises, and upon request or recommendation by their clinician, a referral for specialised hand physiotherapy or occupational therapy. An overview of simple exercises for the three main motions in the wrist (palmar/dorsal flexion, pronation/supination and radial/ulnar deviation) is provided on paper, together with an explanation of exercises by the clinician. Patients are instructed to perform controlled movements up to a point that cause a stretch but no pain. Patients are instructed to practise these movements three to five times daily, approximately 10–15 min per session. This regimen has been developed in cooperation with specialised hand and wrist physiotherapists and occupational therapists, and is most reflective of current standard practice. All patients referred to physiotherapy are instructed to perform active exercises

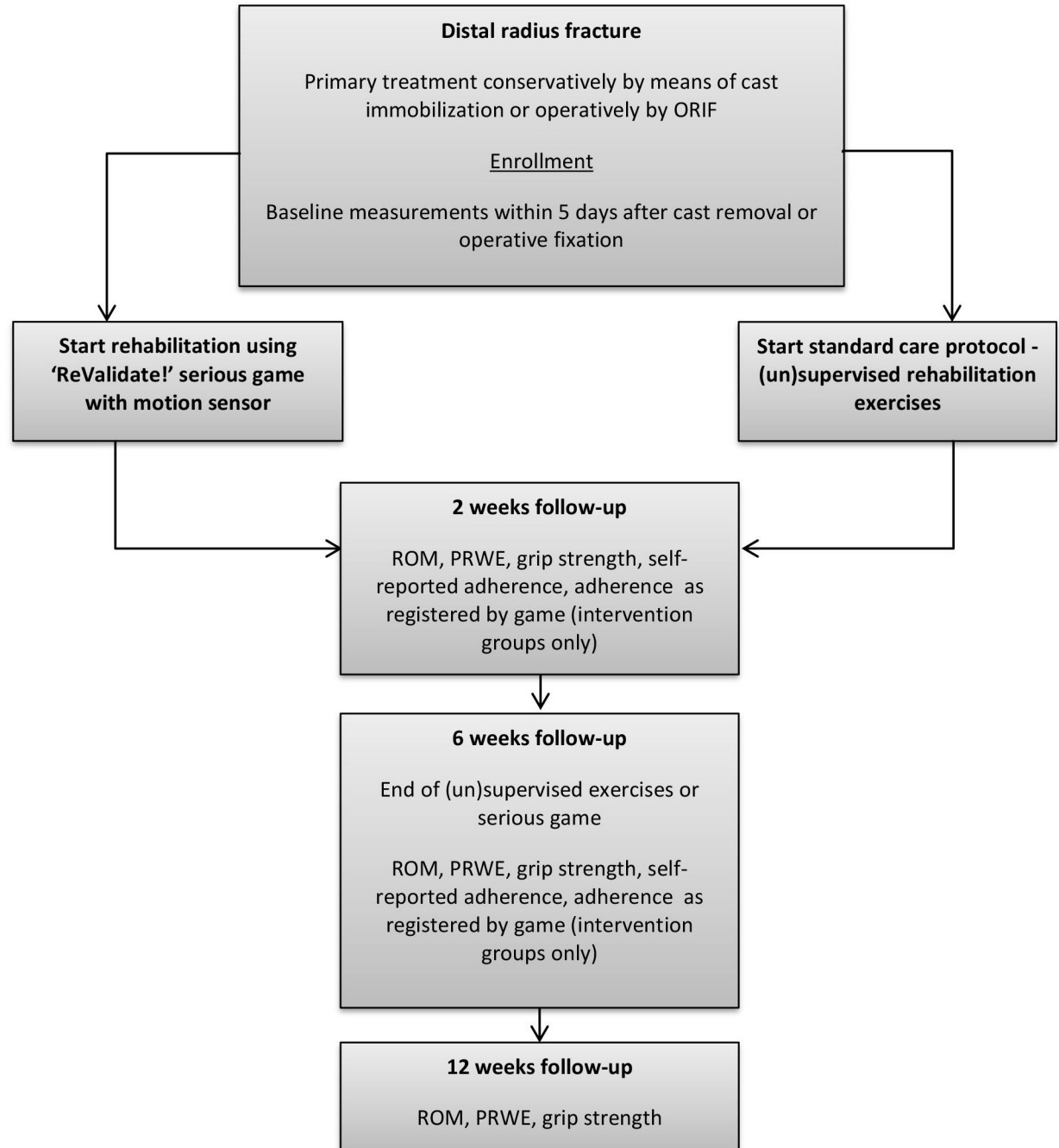

**Figure 3** Schematic timeline of the study showing the randomisation and follow-up planning, as well as the outcomes measured during each visit. ORIF, open reduction internal fixation; PRWE, Patient-Rated Wrist Evaluation; ROM, range of motion.

without load. The number of physiotherapy visits will be registered prospectively.

### Outcome measurements

Standard demographic information including age, sex, fracture type according to the Arbeitsgemeinschaft für Osteosynthesefragen/Orthopaedic Trauma Association fracture classification,[39] and the type of treatment is obtained directly after inclusion. Follow-up visits take place at 2, 6, and 12 weeks postoperatively or after cast removal, for operatively or conservatively treated patients,

respectively. A timeline of the trial is shown in figure 3. The primary outcome measure, which is the validated Patient-Rated Wrist Evaluation (PRWE) questionnaire is completed by patients at all follow-up visits.[40 41] The active ROM is measured in degrees using an analogue universal goniometer. ROM and grip strength are measured in both hands, where grip strength is measured in kilograms, using an analogue hand dynamometer (Baseline, Fabrication Enterprises, White Plains, New York, USA). The maximum value of two measurements with 5 min

rest in between is used for analysis. All measurements are performed by a clinician blinded to group allocation that has been specifically trained for this trial, and is experienced in taking ROM and strength measurements in daily practice.

Patients in both study arms complete a questionnaire, that was specifically developed for this trial, on their self-efficacy, experiences and perceived effectiveness of their exercise programme. Pain scores on a Visual Analogue Scale are recorded, and patients are asked about the frequency, if any, of visits to a physiotherapist or occupational therapist. In addition, partial or complete return to work, or to normal functioning if patients were currently unemployed, is registered. Patients are reminded by the researcher to fill out their exercise diary at every follow-up visit and are asked to return their completed exercise diary at 6 weeks of follow-up. In the intervention group, the data on exercise frequency and duration, as well as ROM, will also be retrieved from the serious game. The frequency reported by patients in this group will be compared with the data from the game application, which stores all exercise data including frequency, duration and the ROM measured.

## Sample size calculation

The primary outcome is the PRWE questionnaire outcome score after 6 weeks of treatment. This questionnaire has been validated previously,[40] and has a minimal clinically important difference of 11.5, with 1.5 for the pain subscale and 10 for the function subscale.[41]

An elaborate and important Cochrane review on the effect of physiotherapy after distal radius fractures reports only three studies comparing a regular physiotherapy regimen with an unsupervised home-exercise programme.[15] These comparisons are relevant here, since the use of this serious game can be compared with an unsupervised home-exercise programme. Only one of these three trials reports patient-rated functional outcomes in addition to ROM or grip strength values.[18] This study looks into the effects of a progressive home-exercise programme, compared with the effect of 12 physiotherapy sessions over a 6-week period in patients with an operatively treated distal radius fracture. They found a positive effect in favour of the home-exercise programme, with PRWE scores of 18.5 (SD 15.9) for the home-exercise group, compared with 36.1 (SD 13.9) in the physiotherapy group after 6 weeks.[18]

As a result, using the SD of 15.9, for a two-sided t-test in two groups with an alpha of 0.05, a power of 90% (1–beta=0.9), an allocation ratio of 1:1, this leads to necessary group sizes of 42 patients per group to detect a clinically relevant difference of 11.5 points at 6 weeks. With a 10% dropout rate, a total of 92 patients are needed for this trial.

## Data analysis

The primary outcome is the PRWE score at 6 weeks. This outcome is represented as the change in outcome scores from T0 to T2 (delta, T2–T0). An intention-to-treat analysis is performed, and any missing data are imputed. Data will be analysed using a two-sided t-test or the Mann-Whitney U test, depending on normality of the distribution.

For the secondary continuous outcomes (PRWE score at 2 and 12 weeks, ROM, grip strength, pain scores), a repeated measures analysis of variance will be performed, where group assignment (game group vs control group) is the between-subjects factor, and time of visit (baseline, 2, 6 or 12 weeks follow-up) is the within-subjects factor. In case of a non-normal distribution, generalised estimating equations will be used. An intention-to-treat analysis will be performed, and any missing data will be imputed using multiple imputation. Other (secondary) functional outcomes are analysed as a percentage of the unaffected wrist. Bonferroni correction is applied to all secondary outcomes to correct for multiple tests. Therefore, p values of <0.0045 will be considered statistically significant.

Baseline data and secondary outcomes include categorical data such as fracture type and treatment type. These data will be analysed using the $X^2$ test, and outcomes will be presented as the differences in frequencies. P values of <0.05 are considered statistically significant. Treatment adherence is measured using a self-reported diary. The number of exercise sessions and the total duration of exercising are continuous variables that will undergo quantitative and qualitative synthesis using a two-sided t-test or a Mann-Whitney U test as appropriate.

The diaries that are handed in after the trial period are analysed for the percentage of days the diary is completed and the total estimated time of exercising. In the intervention group, these data are also compared with the total time of exercising as registered within the game. In addition, should not all diaries be completed, the number of diaries handed in after the study period is registered. These data are then analysed using the $X^2$ test, and outcomes will be presented as the difference in frequencies.

## Patient and public involvement

Patients were involved during development, playtesting and previous validation of the game. No patients were involved in the design, recruitment or conduct of this study. Results of the study will be published on the trial website, which is available publicly. The burden of the intervention has been evaluated by patients taking part in a specialised rehabilitation programme at a hand and wrist physiotherapy practice. They considered playing the game to be a comparable burden with their prescribed exercises.

## ETHICS AND DISSEMINATION

This study will be conducted according to the principles of the Declaration of Helsinki (64th World Medical Association (WMA) General Assembly, Fortaleza, Brazil, October 2013) and in accordance with the Medical

Research Involving Human Subjects Act (WMO). The dataset will be made available and results of our study will be published unreservedly through academic journals, (international) conferences and popular media.

The medical risks of this trial are considered to be low, since treatment of both the intervention and the control group complies with national guidelines for standard treatment of distal radius fractures. Participation, or the choice to not participate, will not have any effect on any part of patient treatment or on the quality of medical care for patients.

Storing and processing of all patient data occurs in compliance with the General Data Protection Act, and a data processing agreement has been established between the app manufacturer and the hospital leading the trial. Digital data gathered by the individual mobile applications have been assessed using a Data Privacy Impact Assessment, and approved of by the hospitals' data privacy officer. All digital data are stored on a computer server located within the highly secured hospital network and can only be accessed by the researchers after logging in with their personal account. The compliance with these rules ensures that privacy risks are minimised.

All data regarding the trial subjects are stored in a secure location in a locked cabinet that can only be accessed by study personnel. Data are anonymised and stored according to subject number. A linking log is stored separately from the data. All data, both digital and hardcopy, are stored for 15 years after trial completion according to WMO regulations.

## DISCUSSION

There is a rapid increase in the use of technological applications for healthcare support and patient self-management.[33 42 43] Clinical research on the effectiveness of these innovative applications is still scarce, however.[32] 'Wearables' show potential as monitoring devices, since they are non-obtrusive and can monitor patients over longer periods of time.[44] In addition, serious games have shown their own potential in education, improving both motivation and learning outcomes,[45 46] and have also shown to be promising tools to increase self-efficacy; an important factor contributing to treatment adherence and physiotherapy outcomes.[28 30]

Readily available computer games using body motion for control, for example, the Nintendo Wii, have already been researched in clinical settings as treatment support tools.[33] Though the issue with off-the-shelf games is that they are not designed as a medical device, hence lack proper validation for medical use. They may increase activity and motivation in a rehabilitation process, yet have been designed for entertainment purposes.[38] Furthermore, these games usually require larger consoles that limit the patient in performing the exercises anywhere they want.

To the authors' knowledge, this study is the first study focusing on a validated, wearable-controlled serious game designed to act as a medical support tool for wrist rehabilitation. This game and similar interventions may decrease the ever-growing burden wrist injuries pose to patients and to society, and can make validated wrist exercise therapy easily accessible for anyone. This study will contribute to the advancements of serious games for traumatic and non-traumatic wrist and other injuries, and may pave the way for future development and validation of wearable-controlled rehabilitation games.

**Author affiliations**
[1]Department of Surgery, Amsterdam Movement Sciences, Amsterdam UMC Location AMC, Amsterdam, The Netherlands
[2]Department of Surgery, Amsterdam UMC Location AMC, Amsterdam, The Netherlands
[3]Department of Plastic, Reconstructive and Hand Surgery, Amsterdam UMC Location AMC, Amsterdam, The Netherlands
[4]Department of Trauma Surgery, Onze Lieve Vrouwe Gasthuis, Amsterdam, The Netherlands
[5]Department of Surgery, Amsterdam Gastroenterology and Metabolism, Amsterdam UMC Location AMC, Amsterdam, The Netherlands

**Acknowledgements** The authors acknowledge all volunteers and patient advisers, physiotherapists and hand therapists who have collaborated during the development of the game. They especially thank A J Videler of the Hand and Wrist Centre, Amsterdam, and C J M Nooij of the Department of Rehabilitation, Amsterdam UMC, location Academic Medical Centre. They also thank Motek Medical and Virtual Play for the collaborative development of the game. Furthermore, they thank all supporting physicians, specialised nurses and other nurses, and outpatient clinic staff from all collaborating hospitals who made the realisation of this trial possible.

**Collaborators** Professor P Kloen, Amsterdam UMC, location Academic Medical Centre, Amsterdam, the Netherlands; H W Bolhuis, Gelre ziekenhuizen, Apeldoorn, the Netherlands; S R de Wild, Gelre Ziekenhuizen, Apeldoorn, the Netherlands; T Herklots, Gelre Ziekenhuizen, Apeldoorn, the Netherlands; J Baaij, Gelre Ziekenhuizen, Apeldoorn, the Netherlands; J Winkelhagen, Dijklanderziekenhuis, Hoorn, the Netherlands; R Buijsman, Amsterdam UMC, location Academic Medical Centre, Amsterdam, the Netherlands; K T van Hamersveld, Dijklanderziekenhuis, Hoorn, the Netherlands; B A van Dijkman, Flevoziekenhuis, Almere, the Netherlands; M Jansen, Amsterdam UMC, location Academic Medical Centre, Amsterdam, the Netherlands; E Z Barsom, Amsterdam UMC, location Academic Medical Centre, Amsterdam, the Netherlands; S L van der Storm, Amsterdam UMC, location Academic Medical Centre, Amsterdam, the Netherlands; N van Oorschot, Amsterdam UMC, location Academic Medical Centre, Amsterdam, the Netherlands; M J van der Pols, Amsterdam UMC, location Academic Medical Centre, Amsterdam, the Netherlands.

**Contributors** All authors have contributed to the design of this trial protocol. MPS and JCG initiated the project. The protocol was drafted by HAWM and revised by MCO, MG, MPS, SvD and JCG. SvD provided statistical and methodological support. All authors have read and approved the final manuscript. The ReValidate! collaborative study group consists of local investigators who are responsible for trial execution, patient inclusion and data collection. They have all read and approved the final manuscript.

**Funding** The ReValidate! project received funding from the Growing Games Program (Dutch Game Garden, iMMovator, Dutch Games Association and the Economic Board Utrecht), grant number CB00018, and by a grant from the CZ Fund, the Netherlands.

**Disclaimer** The sponsor had no role of any sort in the study design.

**Competing interests** None declared.

**Patient and public involvement** Patients and/or the public were involved in the design, or conduct, or reporting, or dissemination plans of this research. Refer to the Methods section for further details.

**Patient consent for publication** Not required.

**Provenance and peer review** Not commissioned; externally peer reviewed.

**ORCID iDs**
Henriëtte A W Meijer http://orcid.org/0000-0002-2342-9446
Marlies P Schijven http://orcid.org/0000-0001-7013-0116

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
