## [Reviewer comments · BMJ Open]

ARTICLE DETAILS

TITLE (PROVISIONAL)	A serious game versus standard care for rehabilitation after distal radius fractures: a protocol for a multicentre randomised controlled trial
AUTHORS	Meijer, Henriëtte; Graafland, Maurits; Obdeijn, Miryam; van Dieren, Susan; Goslings, J. Carel; Schijven, Marlies; ReValidate!, Collaborative research group

VERSION 1 – REVIEW

REVIEWER	YAVUZ YAKUT Hasan Kalyoncu University Gaziantep Turkey
REVIEW RETURNED	12-Aug-2020

GENERAL COMMENTS	In the study, the treatment that patients will receive in the control group should be written in more detail. Physiotherapy or occupational therapy are not the same and can differ. In terms of objectivity of the study, issues such as treatment protocols to be given, treatment durations should be clearly stated.
--

REVIEWER	Rachel kizony University of Haifa, Israel
REVIEW RETURNED	24-Aug-2020

GENERAL COMMENTS	The paper describes a protocol for an RCT that aims to assess the effectiveness of wearable device & serious game application in rehabilitation of wrist injuries. The proposed study is relevant and important to the field of hand rehabilitation and overall the study was carefully planned and the authors provided most of the needed information. Below are some concerns and comments to be addressed by the authors: Introduction: The references of the first paragraph should be updated as most of them are more than a decade ago, and prevalence and incidence of an injury should be more updated. On page 6 (line 37) the authors refer to surgical guidelines; they should state of which country (it seems that from the Netherlands only), and add information about guidelines from other countries. In addition, the reference regarding referral to PT is from 2010 and should be updated. On page 6 (line 53) the authors stated: "Patients with a low self-efficacy will also perceive more barriers to treatment adherence.[26]" – according to the reference this is mainly when referring to home exercise and this should be emphasized. The rationale to the study and specifically how self-efficacy relates to
--

	the methods used in the study, should be written more clearly. Methods: Criteria for exclusion are not clearly stated and are not supported by cut-offs scores of clinical tests. For example, on page 8 (line 32) "history of loss of function due to injury or illness in either hand or wrist"- what do you mean loss of function? What about mild impairment such as early stages of osteoarthritis or Parkinson's disease? How are you going to measure it? There are questionnaires that can be used. Same for mental impairment- how will this be measured? In addition, is there an upper age limit 80, 90 years old? Who will monitor the diaries the participants are writing to make sure they fill them in. The participants must own IPAD or iPhone- this might cause a bias of the sample. Is this the most common type of smartphone/tablets used in the place where the study will be carried out? It is not clear how the participants that use their phone will see the game while the phone is strapped on their forearm. Not clear how much contact the participants will have with clinicians after given the initial instructions for using the game. Also not clear who will instruct the participants` surgeons, OTs, PTs? The authors stated that "The patient plays one level 3-5 times per day" – what is the duration of the exercise? Are all the levels the same? How it is determined whether the participant is ready to proceed to a different level (in terms of his ROM/pain etc.)? Are referrals to PT and OT will be handled the same in both groups? It is not clear from what is written on page 11. The authors should specify what is their primary outcome measure when they describe the measurements. Add a reference and psychometric information about PRWE on page 11. What ROM will be measured and by whom (PT/OT)? Are assessors blind to group allocation; this information is not stated clearly in the paper itself. It is a major limitation of the study and I do not understand why assessors cannot be blind to group allocation. It reduces that internal validity of the study. There are no details about this: questionnaire on their experiences and perceived effectiveness of practice. Is this developed for the current study? What about self-efficacy? This is a major part of the study's rationale and could be a confounder variable, but it is not measured. How will the change score of PWRE be calculated? As a simple delta between T0 and T2 or relative to T0? In data analysis section what is the difference between visit (baseline, 2, 6 and 12 weeks) and time of visit (baseline, 2, 6 or 12 weeks follow up)? What will be the alternative for ANOVA RM if data is not distributed normally? Page 13 lines 32-41 are not clear: for example, the authors stated that the p value is below 0.05 and then write that Bonferroni corrections will be performed- for which variables? What will be the p value after the corrections? What are the categorical variables? This is not clear as well :” The number of diaries handed in after the study period are analysed using the Chi-square test, and outcomes will be presented as the difference in frequencies.” SPIRIT table: Blinding is not described on page 7 as written in the table. Data management is only partially described. Confidentiality is partially discussed in the manuscript but not referred to in the table.
--	--

REVIEWER	Emily Lalone Western University
REVIEW RETURNED	26-Aug-2020

GENERAL COMMENTS	-can you provide more detail regarding the types of centers involved in this study? hand specialized? tertiary care? how many fractures would your clinic see a year? how long will this study take? what sort of recruitment track record do these centers have? -how many patients were not eligible to participate because they didn't have a smart phone? can you rationalize why you didnt just give them a smartphone (with no sim card?) so they can participate? can you comment on the bias you are introducing? -can you comment or provide more detail on the game? -you indicate that for ORIF, patients were told to exercise withing 3-5 days. even at our own institution, some surgeons require 6 weeks and others 3-5 days, so how consistent is this? are you changing care? -what are you going to do in the analysis to account for these methodological differences in sensor type -can you explain your rationale for 6 weeks of exercise -what evidence do you have that the types of rom for the game and the PT indicated therapy are similar. we need more information to show that they are similar such that the only difference is adherence and maybe how they do the exercises but are the underlying PT strategies the same? -do the patients at the clinic who receive PT there have any follow-ups? -how are you going to control the fact that some patients want both PT types?? -i dont see a timeline for study schedule.
---

VERSION 1 – AUTHOR RESPONSE

Itemized list of revisions

1. The 'Strengths and limitations' section has been revised according to your suggestions (page 3, clean copy).
2. As suggested by the editor and by reviewer 2, the inclusion and exclusion criteria have been explained further. The full inclusion and exclusion criteria have been added in Table 1 (page 7, clean copy). This table also shows the rationale for the exclusion criteria. As patient-rated outcome scores are used, it is an exclusion criteria if patients believe their hand/wrist was impaired before the wrist fracture (i.e. clinically relevant loss of function or pain). There are no upper age limits, but patients need to be able to fill out questionnaires and use a smartphone or tablet (subjective judgment by the patients, yet clinically relevant).
3. An English translation of the patient information and consent form has been added as a supplementary file (Supplementary material).
4. A section 'Patient and public involvement' has been added (page 14, clean copy) and the collaborating parties are thanked accordingly in the 'Acknowledgments' section.
5. As rightly stated by reviewer 1 and reviewer 2, the treatment protocols for exercise therapy (including physiotherapy and occupational therapy) as well as the clinician providing exercise

explanations were unclear. We have added explained exercise schedules, durations of exercising, and we have also elaborated on the way these are explained and provided to patients. All referrals are handled in the same way and instructions to physiotherapists and occupational therapists are to support in active exercises without loading the hand or wrist (pages 8, 10-11, clean copy).

6. The references in the 'Introduction' section have been updated as suggested by reviewer 2.

Thank you for this suggestion, we have updated these references as far as was possible with the current/latest known literature on this topic (page 4, clean copy).

7. As stated by reviewer 2, referenced surgical guidelines need to be explained. The most recent versions of the Dutch, German, British and American guidelines are referenced. The manuscript also references the AO Surgery guide, which is a world-renowned guideline (page 4, clean copy).

8. As correctly stated by reviewer 2, self-efficacy as explained in the referenced studies mainly refers to home-based exercises. Thank you for your feedback, we have corrected this (page 5, clean copy).

9. According to the feedback provided by reviewer 2, the rationale of the study and its relation to self-efficacy has been explained in more detail in the 'Introduction' section and specifically in the hypothesis (page 5-6, clean copy).

10. As suggested by reviewer 2, the way diaries are monitored is explained further (page 12, clean copy).

11. Reviewer 2 mentions it is unclear how participants will see the smartphone while playing the game. The setup of the sensor placement using the smartphone and the motion sensor is now explained in more detail, and is also shown in Figure 1 (page 8-9).

12. In addition to a more elaborate explanation of exercise schedules (as commented in item 4 of this list), the gameplay has also been specified. We would like to thank reviewer 2 and reviewer 3 for their suggestions. The gameplay, duration and various levels are explained on page 9-10 (clean copy).

13. The primary outcome measurement has been stated more clearly. In addition, references to reports of the previous validation of the Patient-Rated Wrist Evaluation (PRWE) have been added (page 11, clean copy).

14. How range of motion is measured, and which motions are measured is explained on page 11 (clean copy).

15. Reviewer 2 rightly states that the description of blinding was unclear. This has been explained in the 'Outcome measurements' section on page 11 (clean copy).

16. We would like to thank reviewer 2 for the comments on the questionnaire evaluating motivation, patients' perceived effectiveness of exercising, and self-efficacy. This questionnaire has been specifically developed for this study and has been explained in more detail now in the 'Outcome measurements' section on page 11 (clean copy).

17. The change in PRWE-score will be calculated as delta ($T_2 - T_0$), this has been added in the first paragraph of the 'Data analysis' section on page 13 (clean copy).

18. Reviewer 2 asks what the difference is between 'visit' and 'time of visit'. Our apologies for the confusion here. These refer to the same thing. The error has been corrected (page 13, clean copy).

19. Thanks to reviewer 2 for correctly pointing out there needed to be a plan for analysis of non-normally distributed data. Should data not be distributed normally, Generalised Estimating Equations will be used, as specified on page 13 (clean copy).

20. Reviewer 2 states that the statistical analyses have not been explained clearly. This has now been corrected. The application of Bonferroni correction and the corresponding P-value has been elaborated on (page 13, clean copy). A more thorough explanation of how the diaries are analysed has been added on page 13-14 (clean copy). Thank you for these suggestions.

21. The comments made by reviewer 2 on the SPIRIT table have been corrected, and the SPIRIT table has been updated in total to correspond to the revised version of the manuscript. Data management strategies have been elaborated on in the 'Ethics and dissemination' section

(page 15, clean copy).

22. As remarked by reviewer 3, it was unclear when patients had to start exercising after immobilization. This has been clarified in the manuscript. There is no change in regular care; the protocol merely allows for the normal variation that is seen in the participating hospitals in regular care (page 8, clean copy).

23. Reviewer 3 questions if there is a methodological difference between the two types of motion sensor: there is no difference. Though patients use a different sensor, the placement of sensors is exactly the same, as are the game, the levels of the game and the gameplay and game control. Therefore, the only difference to be accounted for may be the screen size patients use to play the game on. As it is expected that this will not lead to relevant differences in outcome, these two sensor types do not lead to separate groups. This has been clarified on page 9 (clean copy).

24. Reviewer 3 asks for more evidence that the game and physiotherapy exercises are similar. This has been added in the manuscript (page 10, clean copy). Furthermore, the previously published manuscript explaining the face validity and content validity of the game, as well as the validity of its exercise durations has been added as a reference (reference 38).

25. The questions posed by reviewer 3 on physiotherapy and follow up visits have been answered on page 10 and 11 (clean copy). The number of physiotherapy visits will be registered prospectively, and are handled the same in both groups. Patients randomised to the game group but that visit a physiotherapist regardless, will be registered as 'protocol violation', but will still be analysed in the intention-to-treat analysis. The total and average number of physiotherapy visits in both groups will be analysed and will be represented in the final manuscript (page 10-11, clean copy).

26. A schematic representation of the study timeline has been added in Figure 3 (page 11, clean copy).